

# The skyrmion-bubble transition in a ferromagnetic thin film

Anne Bernand-Mantel$^\star$, Lorenzo Camosi, Alexis Wartelle, Nicolas Rougemaille,
Michaël Darques and Laurent Ranno

Université Grenoble Alpes, CNRS, Institut Néel, 25 Avenue des Martyrs, 38042 Grenoble,
France

$\star$ anne.bernand-mantel@neel.cnrs.fr

## Abstract

Magnetic skyrmions and bubbles, observed in ferromagnetic thin films with perpendicular magnetic anisotropy, are topological solitons which differ by their characteristic size and the balance in the energies at the origin of their stabilisation. However, these two spin textures have the same topology and a continuous transformation between them is allowed. In the present work, we derive an analytical model to explore the skyrmion-bubble transition. We evidence a region in the parameter space where both topological soliton solutions coexist and close to which transformations between skyrmion and bubbles are observed as a function of the magnetic field. Above a critical point, at which the energy barrier separating both solutions vanishes, only one topological soliton solution remains, which size can be continuously tuned from micrometer to nanometer with applied magnetic field.

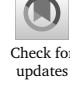

# 1  Introduction

Skyrmions, are topological solitons which present particle-like properties: they have quantized topological charges, interact via attractive and repulsive forces, and can condense into ordered phases. The concept of skyrmions has spread over various branches of physics [1] including condensed matter, as for example in the case of liquid crystals [2], quantum Hall magnets [3, 4] and Bose-Einstein condensates [5]. In ferromagnets, skyrmions were originally called two-dimensional (2D) topological solitons or magnetic vortices and their existence has been predicted in isotropic ferromagnets [6], uniaxial ferromagnets [7–9] and non-centrosymetric magnets [8,10]. Some early indirect experimental evidences of their existence have been obtained in quasi-2D antiferromagnets [11–13]. More recently indirect [14] and direct observations of skyrmion have been reported in chiral magnets [15] and ultrathin layers of conventional transition-metal-based ferromagnets in contact with heavy metals [16, 17].
The nanoscale size and non-trivial topology of skyrmions make them particularly attractive for information technologies. The idea of a skyrmion-based memory was already discussed in the 80's [18] at the time when the research on magnetic bubbles was at its apogee. Magnetic bubbles are cylindrical magnetic domains which appear in ferromagnetic films with perpendicular magnetic anisotropy when the demagnetising energy compensates the domain wall (DW) energy [19]. Memories based on magnetic bubbles displaced by a rotating magnetic field were commercialised in the 90's but were progressively replaced by transistor based memories and hard disk drives due to their higher capacity and lower cost. The recent observations of skyrmions at room temperature (RT) and their fast displacement with low electrical currents [20–22] has triggered a revival of the quest for a memory based on topological solitons, taking the form of a skyrmion racetrack memory [23–25].
Magnetic bubbles and skyrmions are close relatives as they can share the same topology [26]. However, their characteristic sizes differ and while classical bubbles present a long lifetime at RT, much shorter lifetimes were found for nanometer-sized skyrmions in recent experimen-

tal [27,28] and theoretical works [29–31]. The case of intermediate-size solitons is more favorable, as stable RT topological solitons with sizes of a few hundred to a few tens of nanometers have been reported in multilayers [32] and even in a single ferromagnetic layer [33,34]. These topological solitons are sometimes called skyrmion bubbles when the demagnetising energy plays a role in their stabilization. In this context, the necessity to clarify whether a fundamental difference exist between skyrmions and bubbles appears. In previous works, the difference between magnetic bubbles with a large number of collinear spins in their center and skyrmions with a compact core has been described [35, 36]. The dynamic transformation between a magnetic bubble and a skyrmion using a magnetic tip has also been demonstrated [37]. As minimising the total energy of the soliton is numerically expensive, we derive in the present work an analytical topological soliton model in order to build a skyrmion-bubble phase diagram and obtain a better physical insight of the differences between skyrmions and bubbles. This allows us to calculate skyrmions and bubbles equilibrium solutions out of a single model from nanometer to micrometer radius and demonstrate the existence of transitions between them as a function of magnetic field.

## 2 Topological soliton model

The model is developed in view of describing isolated skyrmions and bubbles spin textures in an infinite ferromagnetic thin film. As both spin textures have the same topology, we will use the more general name topological soliton to discuss the model solutions in the first place and later specify what type of topological soliton solution presents the characteristic of a magnetic skyrmion or bubble. The system we consider is a 1 nm-thick ferromagnetic thin film with an easy axis perpendicular to the plane. The material is described by its thickness $t$, its spontaneous magnetisation $M_s$, its exchange constant $A_{ex.}$, its volume magnetocrystalline anisotropy $K_u$ and its micromagnetic Dzyaloshinskii-Moriya interaction (DMI) parameter $D$. The magnetocrystalline anisotropy and the DMI are written as volume-related quantities $K_u$ and $D$. The soliton is assumed to have a cylindrical symmetry and cylindrical coordinates are used, where $\rho$ is the radial distance and the $z-$axis is perpendicular to the film. We restrict ourselves to $D$ values sufficiently large so that DWs will have a Néel nature (based on Ref. [39]) and consequently we also assume a Néel-type for the topological soliton (represented in Figure 1b). In these conditions, the magnetisation direction is defined only by the angle $\theta$ between $\mathbf{M}$ and the z−axis ($\theta = 0$ is up, $\theta = \pi$ is down). A magnetic field $\mu_0 H$ may be applied, for which the positive direction is in the $+z$ direction i.e. antiparallel to the magnetisation in the soliton center (see Figure 1a and b). The film is thinner than the exchange length $l_{ex.} = \sqrt{2A_{ex.}/\mu_0 M_s^2}$ so we assume that magnetisation does not depend on z. In the following, we will present the energy functional containing all the energy terms required to describe the topological soliton energy (Section 2.1). Then, in a first step, we will use the local approximation for the demagnetising energy, generally used for skyrmions [26,40–42], to calculate the topological soliton profile and energy (Section 2.2 and Section 2.3). In a second step, we will show that analytical expressions of the energy terms can be used to calculate the skyrmion solution (Section 2.4) and we will extend the model by taking into account the long range surface demagnetising effect which is at the origin of the bubble solution stabilisation (Section 2.4.4).

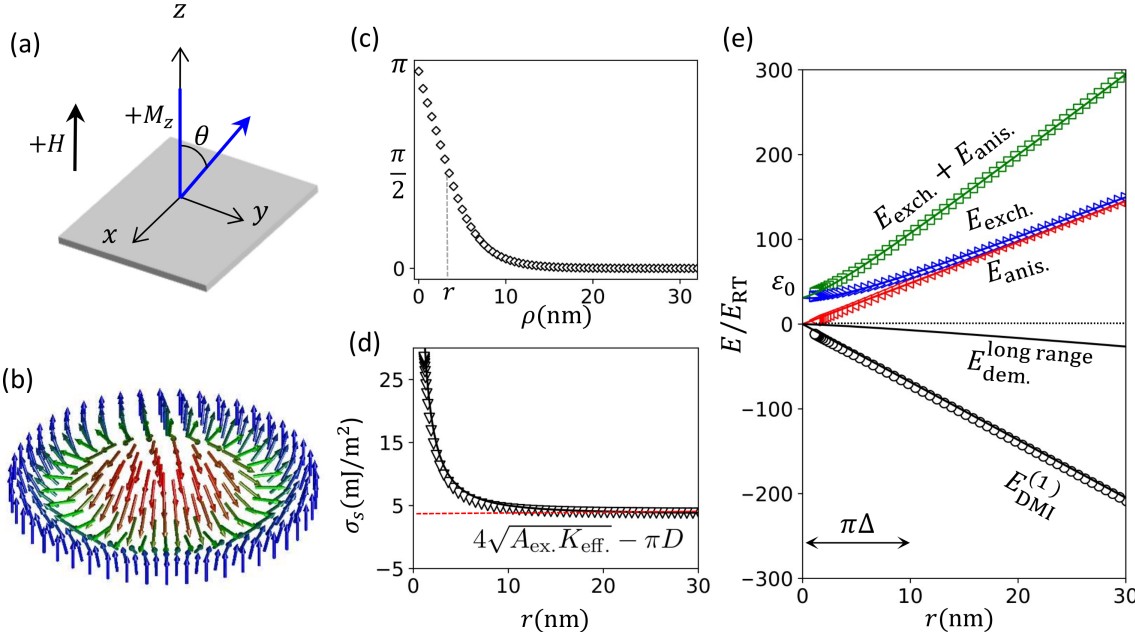

Figure 1: (a) Schematic view of the ferromagnetic thin film with axis orientation. The magnetisation and magnetic field positive directions are indicated. (b) Schematic view of the topological soliton. (c) Energetically minimised topological soliton $\theta(\rho)$ profile as a function of $\rho$ calculated with the parameters $M_s = 1$ MA/m, $K_u = 1.6$ MJ/m$^3$, $A_{ex.} = 10$ pJ/m, $t = 0.5$ nm and $D^{(1)} = 2.8$ mJ/m$^2$. (d) Topological soliton wall energy density $\sigma_s$, defined in Section 2.3, as a function of radius, calculated from Eq. (2) (black triangles) and Eq. (9) (black line). The planar DW energy density $\sigma_w$ is indicated by a dashed line. (e) Energies versus topological soliton radius calculated from Eq. (2) (symbols) and Eq. (9) (lines) with the parameters given in (c) and normalised by the RT thermal energy $E_{RT} = k_B T_{293K}$. The normalised zero radius energy is indicated as $\varepsilon_0 = E_0/E_{RT} \sim 31$. All the programs used to create the data for the figures can be found in a repository [38].

## 2.1 Energy functional

The topological soliton energy $E_s[\theta(\rho)]$ is the sum of 5 terms: exchange energy $E_{exch.}$, anisotropy energy $E_{anis.}$, DMI energy $E_{DMI}$, demagnetising energy $E_{dem.}$ and Zeeman energy $E_{zee.}$:

$$E_s[\theta(\rho)] = 2\pi t \int_0^\infty \left\{ A_{ex.} \left[ \left( \frac{d\theta}{d\rho} \right)^2 + \frac{\sin^2\theta}{\rho^2} \right] - K_u \cos^2\theta \right.$$
$$\left. - D \left[ \frac{d\theta}{d\rho} + \frac{\cos\theta\sin\theta}{\rho} \right] - \mu_0 M_s H \cos\theta + E_{dem.} \right\} \rho \, d\rho. \quad (1)$$

The demagnetising energy can be decomposed into two terms $E_{dem.} = E_{dem.}^{vol.} + E_{dem.}^{surf.}$ corresponding to the contributions from volume and surface magnetic charges. The volume charges appear only inside the topological soliton. The surface charges are present inside the soliton but also in the infinite region with uniform magnetisation. Due to the long range nature of the demagnetising effects, it is analytically impossible and numerically difficult to find the function $\theta(\rho)$ which minimizes the $E_s$ functional. This has been done in the early 70's, in absence of DMI, in the case of magnetic bubbles stabilised by the surface demagnetising energy [43,44] and by Kiselev and co-workers in presence of DMI [35]. In other works, the demagnetising energy is often taken into account only as a local energy contribution [26,40–42] as described in the next Section.

## 2.2 Local approximation for the demagnetising field

We restrict ourselves to the ultrathin film limit where the DW width is much larger than the film thickness and where $E_{\text{dem.}}^{\text{vol.}}$ is negligible compared to $E_{\text{dem.}}^{\text{surf.}}$. Indeed, in a thin film, the demagnetising contribution from the volume charges inside a Néel DW is proportional to the ratio between the film thickness and the DW width [45]. This consideration is also valid for a Néel skyrmion which possesses a $2\pi$ DW cross section. In addition, the film thickness is much smaller than the exchange length and we make a short range uniform magnetisation assumption. In these conditions the surface charge contribution to the demagnetising field is $H_{\text{d}} = -M_{\text{z}}$. We obtain $E_{\text{dem.}}^{\text{surf.}} = \mu_0(-M_{\text{z}}) \cdot (-M_{\text{z}})/2 = +\mu_0 M_{\text{s}}^2 \cos^2\theta/2$. The local surface demagnetising energy becomes equivalent to an anisotropy and the anisotropy constant in Eq. (1) is replaced by an effective anisotropy constant $K_{\text{eff.}} = K_{\text{u}} - K_{\text{d}}$ where $K_{\text{d}} = \mu_0 M_{\text{s}}^2/2$. Considering now the energy difference between the topological soliton and the uniform state we obtain the following Euler Equation [26]:

$$A_{\text{ex.}}\left[\frac{\text{d}^2\theta}{\text{d}\rho^2} + \frac{1}{\rho}\frac{\text{d}\theta}{\text{d}\rho} - \frac{\sin 2\theta}{2\rho^2}\right] + D\frac{\sin^2\theta}{\rho} - K_{\text{eff.}}\frac{\sin 2\theta}{2} - \mu_0 M_s H \sin\theta = 0. \tag{2}$$

## 2.3 Soliton energies versus radius

We solved the Euler-Lagrange Equation (2) using the shooting method starting from an inverse tangent try function and the two boundaries values : $\theta(0) = \pi$ and $\theta(\infty) = 0$. The energetically minimised soliton profile $\theta(\rho)$ is shown in Figure 1c. The topological soliton radius r is defined as the $\rho$ value at which $\theta(\rho) = \pi/2$ ($M_{\text{z}} = 0$). The exchange (blue triangles), anisotropy (red triangles) and DMI energies (black circles) are plotted versus topological soliton radius $r$ in Figure 1e. The Bloch DW width defined as $\pi\Delta = \pi\sqrt{A_{\text{ex.}}/K_{\text{eff.}}}$ is indicated. The DMI and anisotropy energies present a linear variation versus $r$ down to $r < \pi\Delta$. On the contrary, in the same $r$ range, the exchange energy deviates from linearity and tends toward a non-zero constant value. This is due to the curvature $1/\rho$ exchange term in Eq. (1) which is related to the rotation of the spins in the (x,y)-plane: when the topological soliton radius is decreased, the angle between two adjacent spins in this plane increases and the exchange energy increases, leading to a non zero exchange energy limit when $r \to 0$. Consequently the topological soliton wall energy density $\sigma_{\text{s}}$ defined as the sum of the exchange, anisotropy and DMI energy densities deviates from the planar DW energy density $\sigma_{\text{w}} = 4\sqrt{A_{\text{ex.}}K_{\text{eff.}}} - \pi D$ in the low $r$ range as illustrated in Figure 1d. This non-linearity in the exchange term is at the origin of the presence of a local minimum in the total energy in presence of a sufficiently large DMI as we will discuss in Section 3.1. As $E_{\text{anis.}}$, $E_{\text{DMI}}$ and $E_{\text{exch.}}$ show relatively simple $r$ dependences, we will in the following derive analytical expressions to reproduce these dependences.

## 2.4 Analytical model

We now introduce analytical expressions for the 5 different energy terms which constitute the total energy of the topological soliton. The total energy, as well as each energy term is defined as the energy difference between an isolated topological soliton with a down magnetisation in its core and the uniform ferromagnetic state in the up direction.

### 2.4.1 Anisotropy and exchange energies

For $r \gg \pi\Delta$ the sum of exchange and anisotropy energies is proportional to $4\sqrt{A_{\text{ex.}}K_{\text{eff.}}}$ and both energies contribute equally :

$$E_{\text{anis.}} = 2\sqrt{A_{\text{ex.}}K_{\text{eff.}}} \cdot 2\pi r t, \tag{3}$$

$$E_{\text{exch.}} = 2\sqrt{A_{\text{ex.}}K_{\text{eff.}}} \cdot 2\pi r t. \tag{4}$$

For $r < \pi\Delta$, the exchange energy deviates from linearity as discussed in Section 2.3 and we modify this expression by adding a low $r$ correction:

$$E_{\text{exch.}} = 2\sqrt{A_{\text{ex.}}K_{\text{eff.}}} \cdot 2\pi r t + \frac{E_0}{\frac{2r}{\pi\Delta} + 1}, \tag{5}$$

which gives the exact $E_0 = 8\pi A_{\text{ex.}} t$ zero-radius limit for the exchange energy, (Belavin and Polyakov [6], see also [26, 46]) and a zero ($r = 0$) $dE_{\text{exch.}}/dr$ derivative. The analytical expressions of $E_{\text{anis.}}$ and $E_{\text{exch.}}$ as well as their sum appear as lines in Figure 1e. This expression reproduces the exchange energy obtained in Section 2.3 with a 3% maximum error in the full radius range.

### 2.4.2 DMI energy

The DMI energy is proportional to the total $\pi$ rotation of the spins (from the center to the periphery of the topological soliton) and varies linearly with $r$ as observed in Figure 1e. It expresses as:

$$E_{\text{DMI}} = -\pi D \cdot 2\pi r t. \tag{6}$$

We have chosen the rotation chirality which lowers the energy and $D > 0$. $E_{\text{DMI}}$ is negative, thus it favours the expansion of topological solitons.

### 2.4.3 Zeeman energy

We use an approximate expression for the Zeeman energy of the topological soliton which represents the Zeeman energy difference between a film with a uniform $+M_s$ magnetisation, containing a magnetic cylinder of radius $r$ with an opposite uniform $-M_s$ magnetisation, and the Zeeman energy of the uniform $+M_s$ state.

$$E_{\text{Zee}} = 2\mu_0 M_s H \cdot \pi r^2 t. \tag{7}$$

The error associated with this approximation will be discussed in Section 4.3.2.

### 2.4.4 Demagnetising energy

As discussed in Section 2.1 the demagnetising energy, cannot be expressed analytically and approximations have to be used [44]. The local effect of the demagnetising energy in the region where the spin rotates is taken into account using a local approximation and replacing $K_{\text{u}}$ by $K_{\text{eff.}}$ in Eq. (1) as discussed in Section 2.2. This local approximation neglects the long range demagnetising effect which becomes non negligible as the skyrmion radius grows. This long range demagnetising energy contribution is at the origin of classical magnetic bubbles formation [19] and its role in the stabilisation of skyrmions in thin films has been recently shown [34, 46]. We will use a classical expression [47] for the surface demagnetising energy which represents the demagnetising energy difference between a magnetic cylinder of opposite magnetisation included in an uniform ferromagnetic state and the fully uniform ferromagnetic state:

$$E_{\text{dem.}}^{\text{long range}} = -\mu_0 M_s^2 I(d) 2\pi r t^2, \tag{8}$$

where $I(d) = -\frac{2}{3\pi}\left[d^2 + (1-d^2)E(u^2)/u - K(u^2)/u\right]$, $d = 2r/t$, $u^2 = d^2/(1+d^2)$ and where $K(u)$ and $E(u)$ are the complete elliptic integrals of the first and second kind. This formula is a very good approximation to obtain the surface demagnetising energy when the topological

soliton radius is much larger than the DW width $r \gg \pi\Delta$. For $r \sim \pi\Delta$ it leads to an overestimation of this energy as it assumes an abrupt variation of the surface charge density instead of a progressive variation along the DW. The impact of this overestimation is discussed in Section 4.3.1 and Section 4.4.

### 2.4.5 Total energy

We obtain the following analytical expression for the soliton energy with respect to the homogeneous state:

$$
\begin{aligned}
E_{\mathrm{s}} &= E_{\mathrm{exch.}} + E_{\mathrm{anis.}} + E_{\mathrm{DMI}} + E_{\mathrm{Zee}} + E_{\mathrm{dem.}}^{\mathrm{long\ range}} \\
&= \frac{E_0}{\frac{2r}{\pi\Delta} + 1} + 4\sqrt{A_{\mathrm{ex.}} K_{\mathrm{eff.}}} \cdot 2\pi r t - \pi D \cdot 2\pi r t + 2\mu_0 M_s H \cdot \pi r^2 t - \mu_0 M_s^2 I(d) 2\pi r t^2 \qquad (9) \\
&= \sigma_{\mathrm{s}} \cdot 2\pi r t + 2\mu_0 M_s H \cdot \pi r^2 t - \mu_0 M_s^2 I(d) 2\pi r t^2,
\end{aligned}
$$

where $\sigma_{\mathrm{s}}$ is the topological soliton wall energy density defined in Section 2.3. When the radius decreases, it deviates from $\sigma_{\mathrm{w}}$, as shown in Fig. 1d, and the radius dependent correction coming from the curvature of the wall is becoming comparable to $\sigma_{\mathrm{w}}$ itself.

## 3  Topological soliton solutions

Topological soliton solutions are minima in the soliton energy $E_{\mathrm{s}}$ as a function of the topological soliton radius $r$. We call the equilibrium topological soliton radius $r_{\mathrm{s}}$. In order to study the conditions giving rise to these minima, we fix the parameters $A_{\mathrm{ex.}}$, $M_s$, $K_{\mathrm{u}}$ and $t$ and vary $D$. We have checked that changes in the fixed parameters do not modify qualitatively the results presented here but rather shift the main features to different $D$ values. The different energy terms (except the Zeeman energy) are plotted as a function of $r$ (up to 1 μm) in Figure 2a, using the same parameters as in Figure 1, and three different DMI values. The resulting topological soliton energies $E_{\mathrm{s}}$ are shown in Figure 2b. As the different energy terms compensate, a small variation of $D$ is enough to modify the slope of $E_{\mathrm{s}}^{(1)}$, $E_{\mathrm{s}}^{(2)}$ and $E_{\mathrm{s}}^{(3)}$. In the following sections we will describe three type of topological solition solutions, observed for increasing $D$ values.

### 3.1  Skyrmion solutions

As discussed in Section 2.3 and illustrated in Figure 1e, $dE_{\mathrm{exch.}}/dr$ decreases with $r$ in the $r < \pi\Delta$ range. This non linearity may lead to the formation of an energy minimum as observed in Figure 2c. For $r < r_{\mathrm{s}}$ (indicated with an $\alpha$) , the soliton energy variation is dominated by the $dE_{\mathrm{DMI}}/dr$ variation while for $r > r_{\mathrm{s}}$ (indicated with a $\beta$), the $d(E_{\mathrm{exch.}} + E_{\mathrm{anis.}})/dr$ variation is taking over. This soliton solution corresponds to what is usually referred to as a skyrmion: its radius is of the order of a few $\Delta$ or less and it exists down to $H = 0$. The skyrmion radius increases with $D$ and depends weakly on the applied magnetic field as we will see in Section 4.

### 3.2  Bubble solutions and coexistence of skyrmions and bubbles

The second situation occurs when the positive and negative energy terms nearly compensate over a wide $r$ range (see $E_{\mathrm{s}}^{(2)}$ in Figure 2b). In this case, as we can see on Figure 2d, the positive slope of the sum of exchange and anisotropy dominates at intermediate $r$ ($\beta$ part), but $E_{\mathrm{s}}^{(2)}$ decreases again with $r$ at larger $r$ ($\gamma$ part), due to the non-linear increase in $E_{\mathrm{dem.}}^{\mathrm{long\ range}}$.

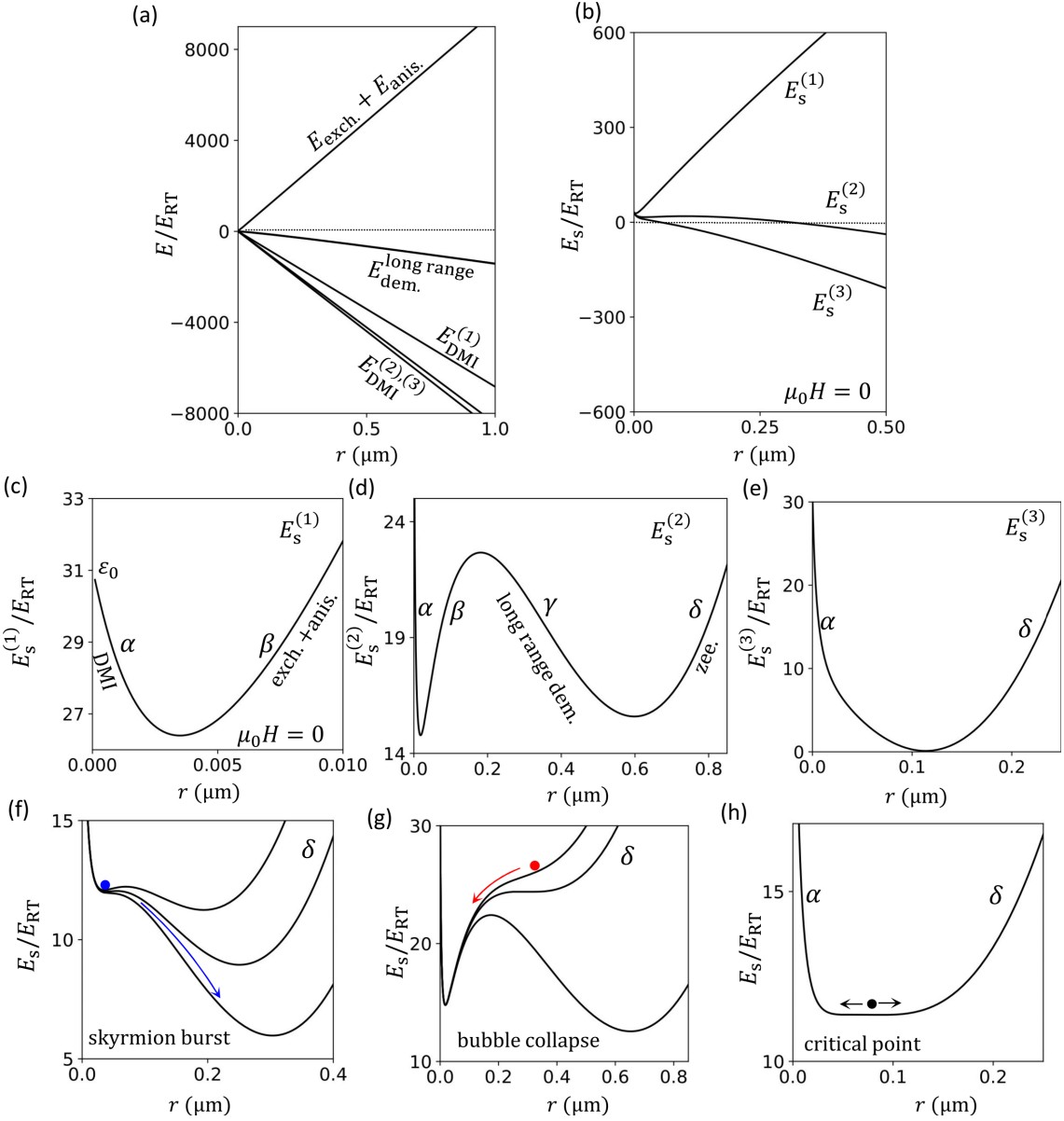

Figure 2: (a) Energies versus topological soliton radius calculated from Eq. (9) with the same parameters as in Figure 1, for zero applied magnetic field and $D^{(1)} = 2.8$ mJ/m$^2$, $D^{(2)} = 3.46$ mJ/m$^2$ and $D^{(3)} = 3.6$ mJ/m$^2$. (b) Topological soliton energies $E_s^{(1)}$, $E_s^{(2)}$ and $E_s^{(3)}$ calculated with the same parameters as in (a). (c), (d) and (e) Topological soliton energies $E_s^{(1)}$, $E_s^{(2)}$ and $E_s^{(3)}$ for applied magnetic fields of respectively $\mu_0 H = 0$ (c), $\mu_0 H = 0.28$ mT (d), $\mu_0 H = 2$ mT (e). The $\alpha$, $\beta$, $\gamma$, and $\delta$ letters are indicating the different parts of the $E_s(r)$ curve showing a monotonous variation. (f), (g) and (h) Topological soliton energies calculated with the same parameters as in (a) and respectively for (f) $D = 3.51$ mJ/m$^2$ and a magnetic field between 0.6 and 0.7 mT for (g) $D = 3.46$ mJ/m$^2$ and a magnetic field between 0.27 and 0.35 mT for (h) $D = 3.52$ mJ/m$^2$ and $\mu_0 H = 0.925$ mT. All the programs used to create the data for the figures can be found in a repository [38].

with $r$, before $E_{\text{zee.}}$ with a $r^2$ variation takes over for larger $r$ ($\delta$ part). Consequently, in presence of magnetic field, two soliton solutions are observed, separated by a local maximum of energy. The lower radius solution is the skyrmion solution described in Section 3.1 and shown in Fig. 2c. The second solution presents the characteristics of what is usually named

a magnetic bubble: it collapses when increasing the magnetic field (see Fig. 2g) and its size diverges at $H = 0$. This coexistence of a skyrmion and a bubble solution was evoked in a pioneering work on skyrmions [8] and described in theoretical works from Kiselev et al. [35] and Büttner et al. [46]. The local maximum of energy creates an energy barrier which is at the origin of hysteretic behaviours in the $M(H)$ loops of magnetic bubble materials [19].

### 3.3  Solutions above a critical $D_{cs}$ value

The third situation occurs when the DMI reaches a critical $D_{cs}$ value above which the local maximum of energy, as observed in Fig. 2d, disappears (see Fig. 2e). In this case, the total energy $E_s^{(3)}$ presents a negative slope at all $r$ in absence of applied magnetic field (Figure 2b). In presence of magnetic field, an energy minimum is restored in $E_s^{(3)}$ as the positive Zeeman energy variation dominates at sufficiently large radius due to its $r^2$ variation ($\delta$ part in Fig. 2e). For increasing magnetic field, this solution can be compressed to very low radius, as it is the case for skyrmions, without encountering a collapse field. When the magnetic field is decreased, the topological soliton radius will increase and diverge at $H = 0$ as it is the case for bubbles. The critical $D_{cs}$ value above which these solutions appears will be further discussed in Section 4.

## 4  Topopogical solitons phase diagram

In Figure 3, we present the evolution of the topological soliton radius $r_s$, calculated with the same $A_{ex.}$, $M_s$, $t$, and $K_u$ parameters as in Figure 1 and Figure 2, as a function of $\mu_0 H$ and $D$. The result is shown for a large range of $D$ and $\mu_0 H$ values (up to 100 mT) in Fig. 3a and for $D$ close to $D_{cs}$ and low fields (close to 1 mT) in Figs. 3d and e. The main features appearing in Fig. 3a, d and e are represented schematically in Figures 3b and f. Vertical cross sections of the diagrams in Figs. 3a and d,e are shown in Figs. 3c and g. Our analytical model allows us to obtain a skyrmion phase diagram similar to the one described by Bogdanov et. al. [48] and Kiselev et. al. [35] (Figure 3b) and to complete it with the bubble solution at low magnetic field (Figure 3f). The asymmetry of the topological soliton phase diagram with respect to the magnetic field comes from the fact that we are describing metastable states compared to the ground state uniformly magnetised in the positive $z$ direction.

### 4.1  Description of the topological soliton phase diagram

We have divided the topological soliton phase diagram in different zones appearing in Figure 3b and f and described in the following.

#### 4.1.1  Single topological soliton: zones 1 and 2

The skyrmion solutions described in Section 3.1 appears in zone 1. These solutions persist down to $H = 0$ and for negative applied magnetic fields (see Figure 3a) and disappear along a blue line visible in Fig. 3b which was defined as the skyrmion bursting line in previous works [35, 48]. This suppression of the skyrmion solution for decreasing magnetic field is illustrated in Figure 2f. The skyrmion bursting line ends at a critical point $(D_{cs}, H_{cs})$ indicated by a blue dot in Fig. 3b at which no local maximum is observed any more in $E_s(r)$. In the works where the long range demagnetising effect is neglected [48], this critical point appears at $H = 0$ and $D_{cw} = 4\sqrt{A_{ex.} K_{eff.}}/\pi$. In our case, the critical topological soliton $D$ value $D_{cs}$ is lowered compared to $D_{cw}$, because we take into account the long range contribution in the demagnetising energy, which stabilizes the topological soliton similarly to the DMI energy term.

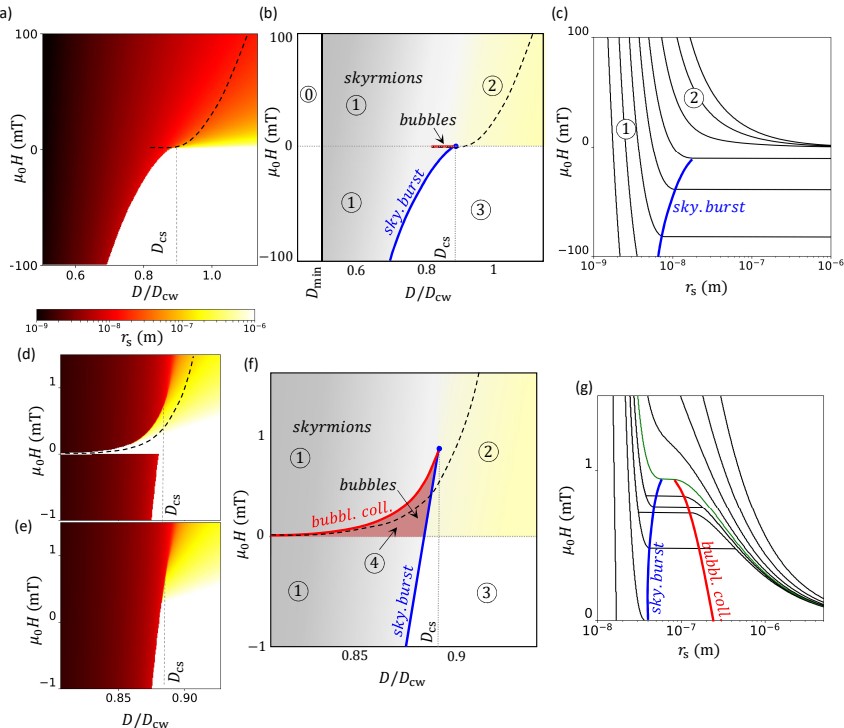

Figure 3: (a) Topological soliton equilibrium radius as a function of applied magnetic field and parameter D, calculated with the same parameters as in Fig. 1. When two solutions coexist the solution corresponding to the larger radius is shown. (d) Zoom of (a) at low magnetic field and close to $D_{cs}$. (e) same as (d) but showing the solution corresponding to the lower radius when two solutions coexist. (b) and (f) Schematic representations of characteristic lines appearing respectively in (a) and (d) and/or defined Section 4.1.1: dashed black line: limit for isolated solitons, red line: bubble collapse line, blue line: skyrmion bursting line. The numbers in circles corresponds to the zones defined in Sections 4.1.1, 4.1.2 and 4.1.3. (c) and (g) Topological soliton equilibrium radius as a function of magnetic field for fixed $D$ values. The D values are ranging from $D/D_{cw} = 0.59$ to 1.13 for (c) and from $D/D_{cw} = 0.87$ to 0.93 for (g). The green line in (g) correspond to $D/D_{cw} = 0.896$, close to $D_{cs}$. All the programs used to create the data and the figures can be found in a repository [38].

The skyrmion radius in zone 1 is always smaller than the radius at the critical point $r_{cs}$ ($\sim$100 nm in our case). In addition the skyrmion radius shows a small susceptibility $dr_s/dH$, except close to the critical skyrmion bursting line (see Fig. 3c). This is due to he fact that the Zeeman energy term is a second order contribution at low $r$ due to its $r^2$ variation.

Zone 2 contains the solutions described in Section 3.3. As observed in Fig. 3c the topological solitons in this zone present a skyrmionic behaviour at high positive magnetic field: low $dr_s/dH$ susceptibility and no collapse field. At low magnetic field their susceptibility is large and increases with decreasing field, as it is the case for bubbles. The dashed line in Figs. 3a,b and d,f is the line at which the isolated topological soliton energy $E_s(r_s)$ becomes negative indicating that the uniform ferromagnetic state is not any more the ground state. Above this line the low energy cost of topological solitons and DWs favours the formation of a topological soliton lattice or a stripe/helical phase as described in the works from Bogdanov et al. [48] and Kiselev et al. [35]. We point out that zone 2 is extending above the critical $D_{cw}$ value. In this zone, for high magnetic fields, the magnetic field compresses $r_s$ down to $r \sim \pi\Delta$ and the

non-linearity in the exchange term causes the soliton energy to increase and become positive again: the isolated soliton solution is restored despite a negative $\sigma_w$ (see Figure 2e). Experimental observations of such metastable isolated topological soliton at high magnetic field can be found in the work from Romming et. al. [17] who reported skyrmions in systems with $D > D_{cw}$ under high magnetic fields of a few Tesla.

### 4.1.2   No topological soliton: zone 0 and 3

Zone 1 starts for $D$ values larger than $D_{min.} = 2\sqrt{A_{ex.}K_{eff.}}/\pi$. Below this value, in zone 0, the DMI energy is too low to compensate the anisotropy energy, the $\alpha$ slope in Figure 2c becomes positive, and there is no energy minimum in $E_s(r)$. This defines a lower DMI value for the formation of a static topological soliton, however, dynamical solitons [49,50], not discussed in the present work, can still be created for example using spin polarised currents [51,52]. In addition, higher order exchange energy terms, not taken into account in the present model can also lead to the stabilisation of skyrmions in absence of DMI [8,9,53,54]. Zone 3 is delimited by the blue skyrmion bursting line at which the skyrmion solution disappear and the $H = 0$ red line. In this zone the negative magnetic field suppresses the isolated metastable state at low $D$. For sufficiently large $D$, at low magnetic field, a topological soliton lattice or a stripe/helical phase is predicted in this zone [26,35,48]).

### 4.1.3   Bubbles solutions and skyrmion and bubble coexistence: zone 4

The region where bubbles appear is defined as zone 4. This zone overlaps with zone 1: a second energy minimum corresponding to the bubble solution appears in $E(r)$ without modifying the skyrmion radius. In Figs. 3d and e we show the topological soliton solution at low magnetic field and for a narrower DMI range compared to Figure 3a. The dashed line in zone 4 correspond to the line at which the bubble energy becomes negative. When two solutions coexist, as discussed in Section 3.2, the larger/smaller solution is shown respectively in Figs. 3d and e. The coexistence zone is delimited by the $H = 0$ line, the bubble collapse red line and the skyrmion bursting blue line. The bubble collapse line and the skyrmion bursting line meet at the critical $(D_{cs}, H_{cs})$ point. The coexistence is also visible in the vertical cross Section of Figs. 3d and e shown in Fig. 3g. When $D$ is decreased, the bubble collapse field decreases and zone 4 vanishes because the topological soliton wall energy becomes to large to be compensated by the long range demagnetising energy.

### 4.2   Transitions close to the critical $(D_{cs}, H_{cs})$ point

In Figure 3g we show the $r_s$ evolution with magnetic field very close to $D_{cs}$. Below $D_{cs}$ the energy barrier which separates the two solutions, visible in Fig. 2d, causes hysteretic behaviors in the bubble-skyrmion transformations. A bubble transform into a skyrmion via collapse for increasing magnetic field and skyrmions abruptly expand when the magnetic field is increased in the negative direction up to the bursting field (see Fig. 2f and g and Fig. 3g). At $D = D_{cs}$ and $H = H_{cs}$ (blue dot in Fig. 3f), the hysteretic behaviour is suppressed (see green line in Fig. 3g). This is due to the suppression of the energy barriers separating the two solutions which leads to a remarkably flat $E_s(r)$ energy profile close to $r_{cs}$ as visible in Fig. 2h. This particularity, which is only observed close to the critical point, is due to an almost perfect compensation, when the radius is varying, of the topological soliton wall energy cost by the surface demagnetising energy gain.

The skyrmion-bubble transition observed here is reminiscent of the critical phenomena observed in the liquid-gaz second order phase transition. Firstly, the transitions occur along lines

that terminate at a critical point $(D_{cs}, H_{cs})$. Secondly, for $D < D_{cs}$ an interval where both solitons coexist is observed similarly to the gas and liquid mixture observed in the temperature versus density plane of the liquid/gas phase diagram. Thirdly, we observe numerically a divergence in the topological soliton compressibility $dr_s/dH$ at the critical point (Figure 3g). In analogy to the opalescence phenomena observed in liquids at the critical point, topological soliton at this point should present remarkable behaviours due to strong thermally activated fluctuations of its radius.

### 4.3 Impact of the model approximations on the critical point position and topological soliton radius

#### 4.3.1 Demagnetising energy

As discussed in Section 4.1.1, the presence of the long range surface demagnetising energy $E_{dem.}^{long\ range}$ shifts the critical $D$ value at which the compensation between positive and negative energy terms occurs. However, as discussed in Section 2.4.4 the expression we use to estimate $E_{dem.}^{long\ range}$ leads to an overestimation of this energy. Consequently, while the $D_{cs}$ calculated with the parameters used in the present work is equal to $0.88 \cdot D_{cw}$, the critical $D_{cs}$ in a real system may be closer to $D_{cw}$. This is confirmed by our micromagnetic simulations presented in Section 4.4 where the $D_{cs}$ value is found to be 8% larger leading to $D_{cs} = 0.95 \cdot D_{cw}$.

#### 4.3.2 Zeeman energy

The error associated with the Zeeman energy analytical expression that we use comes from a non compensation of the up and down $M_z$ component in the region where the spins rotate. When $r \gg \pi\Delta$ this error is negligible ($< 1\%$). For $r = 2\pi\Delta = 20$ nm the error in $E_{Zee}$ is of the order of 5%. For $r = 5$ nm it reaches 30% of $E_{Zee}$. However the impact of this underestimation on the topological soliton radius is limited by the fact that $E_{Zee}$ decreases quadratically with $r$ and that the Zeeman energy and its variations are always one to several orders of magnitude smaller than the total energy value and total energy variations in the $r < \Delta$ range.

### 4.4 Comparison with micromagnetic simulations

We have carried out micromagnetic simulations in order to confirm the predictions obtained using our analytical model and to estimate the impact of our approximations. The simulations have been performed using the Mumax3 open source software from Ghent University [55]. The skyrmion stability under an applied external magnetic field has been computed after discretising the system into orthorhombic cells (finite difference approach). Periodic boundary conditions has been used to limit size effects and the damping coefficient is set to 0.5 to speed up convergence (no magnetization dynamics). The system is a 2048 x 2048 nm$^2$ square box with a mesh size of 0.5 x 0.5 x 0.5 nm$^3$, in which a nanometer sized Néel-type skyrmion is first relaxed. Once the skyrmion is initialized, a 5 mT perpendicular magnetic field is applied and reduced in steps of 0.5 mT. A series of simulations is run for variable D values, ranging from 3.60 to 3.90 mJ/m$^2$. We present the relaxed topological soliton radius as function of applied magnetic field (Figure 4a) and for comparison the equilibrium topological soliton radius obtained analytically using the same parameters except the D values, ranging from 3.40 to 3.60 mJ/m$^2$ (Figure 4b). The results support qualitatively the predictions from the analytical model. In zone 1, the skyrmion solution shows a low $dr_s/dH$ susceptibility. In zone 2, the topological soliton is larger and this susceptibility is increased. In zone 4, the susceptibility varies strongly with decreasing magnetic field. Close to the critical $D$ value $D_{cs}$=3.8 mJ/m$^2$ for Fig. 4a and $D_{cs}$=3.5 mJ/m$^2$ for Fig. 4b we observe a skyrmion burst when the magnetic field is decreased to the critical $H_{cs}$ value. Close to these $D_{cs}$ and $H_{cs}$ values we observe hystretic

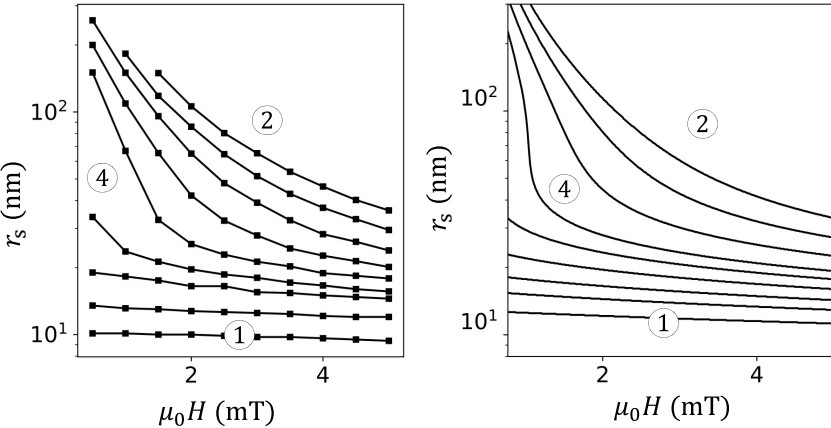

Figure 4: (a) Topological soliton equilibrium radius as a function of applied magnetic field, calculated with Mumax3 using the same $M_s$, $K_u$ and $A_{ex.}$ parameters as in Fig. 1 and $D$ values ranging from 3.60 to 3.90 mJ/m$^2$. (b) Topological soliton equilibrium radius as a function of applied magnetic field calculated with the analytical model (Eq. 9). The parameters are the same as for (a) except for $D$ values ranging from 3.4 to 3.60 mJ/m$^2$.

behaviours in the topological soliton radius variation versus magnetic field in the micromagnetic simulation (not shown here). This behaviour, which reveals a bi-stability, is very similar to both what was reported in a recent micromagnetic study very similar to the one carried out here [56] and to what is predicted by our analytical model (see Figure 3g). To finish we would like to emphasize that such micromagnetic results must be considered with care, especially for bubbles which have a diameter of a fraction of the simulation box. Indeed, when D becomes large ($D > 3.86$ mJ/m$^2$ typically) or when the applied field is small the skyrmion starts to develop a squareness [57]. In addition, the skyrmion starts to feel the edges of the simulation box and confinement effects cannot be neglected any more, explaining the fact that the measured radius is not diverging at very low field.

## 5 The topological soliton skyrmionic factor

In order to estimate the role of the long range surface demagnetising energy in the stabilization of a given topological soliton with radius $r_s$, we introduce the skyrmionic factor $S$ which represent the ratio between the topological soliton wall energy cost and long range demagnetising energy gain:

$$S(r_s) = -\frac{E_{exch.} + E_{anis.} + E_{DMI}}{E_{dem.}^{long\ range}}. \tag{10}$$

The $S$ values corresponding to solutions from Figure 5a are shown in Figure 5b. For $r_s \ll \pi\Delta$ the skyrmionic factor is large: $S \gg 1$ as $E_{dem.}^{long\ range}$ is negligible in this range compared to the topological soliton wall energy cost which tends toward $E_0$ (Figure 1e). On the contrary, large skyrmions with $r_s \gg \pi\Delta$ are only formed when the topological soliton wall energy cost is lower than the energy gain due to long range demagnetising effect which implies $S < 1$. To check this correlation between the size and the $S$ factor of a given topological soliton, we plot the soliton radius as a function of $S$ in Figure 5d. Bubble and skyrmion solutions from zone 4 and 1 are shown respectively in red and black while topological soliton above the critical point (zone 2)

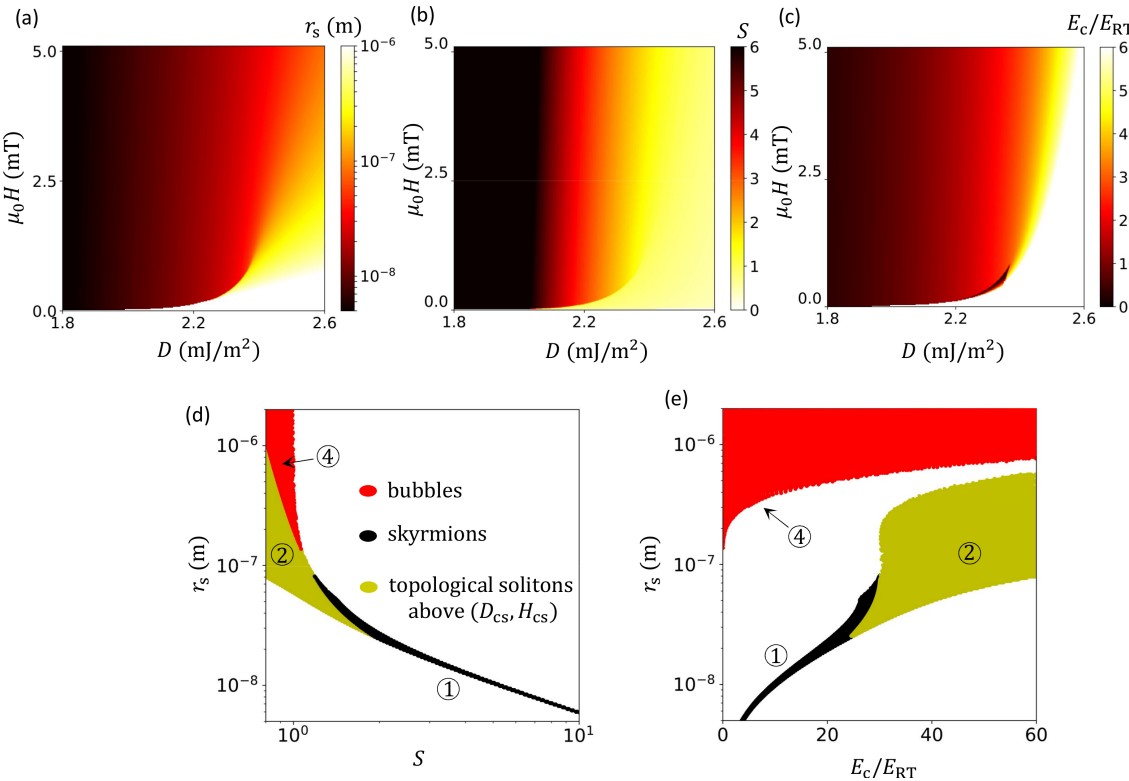

Figure 5: (a),(b) and (c) Topological soliton equilibrium radius (a), $S$ factor (b) and collapse energy barrier $E_c$ normalised by the RT thermal energy $E_{RT} = k_B T_{293K}$ (c), as a function of the micromagnetic DMI, calculated with the parameters $M_s = 1$ MA/m, $K_u = 1$ MJ/m$^3$, $A_{ex.} = 15$ pJ/m, $t = 0.7$ nm. (d) Topological soliton equilibrium radius as a function of the $S$ factor extracted from (a) and (b). (e) Topological soliton equilibrium radius as a function of $E_c/E_{RT}$ extracted from (a) and (c). The numbers in circles corresponds to the zones indicated in the diagrams in Figure 3f. All the programs used to create the data and the figures can be found in a repository [38].

appear in yellow in Figure 5d. Skyrmion and bubble solutions appear in two distinct $r$ ranges, above and below $r_{cs}$. However above the critical point, the radius of topological solitons as well as their S factor can be tuned continuously across $r_{cs}$. Topological solitons with $S$ close to 1 shows a large $r_s$ distribution while the $r_s$ of skyrmions and compact topological solitons at high magnetic field are strongly correlated with $S$. We conclude that the size criteria is relevant to distinguish between skyrmions and bubbles only below the critical $(D_{cs}, H_{cs})$ point.

# 6 Topological solitons stability

The analytical topological soliton model derived here allows us to calculate the energy barrier protecting the solitons from collapse. This collapse energy represents the energy necessary for the topological soliton to annihilate via compression. This gives an estimation of the stability of a topological soliton, keeping in mind that our continuous model may become irrelevant at the atomic scale and that other annihilation mechanism with lower energy paths may exist, in particular in the presence of defects or edges [30]. We define the topological soliton collapse energy barrier as $E_c = E_0 - E_s^{min}$ where $E_s^{min}$ is the local minimum in $E_s(r)$. The bubble col-

lapse energy is defined as $E_c = E_s^{\max} - E_s^{\min}$ where $E_s^{\max}$ is the local maximum. In figure 5c we plot the collapse energies corresponding to solutions from figure 5a divided by $E_{RT} = k_B T_{293K}$. In figure 5e we show the topological soliton equilibrium radius $r_s$ as a function of their collapse barrier $E_c/E_{RT}$ where the topological solitons from zone 2 are represented in yellow while skyrmion and bubbles solutions appear respectively in black and red. The segregation of skyrmion and bubbles in two different $r_s$ ranges, due to the presence of an energy barrier between them in $E_s(r)$, appears again clearly. In addition, bubbles have a strong dispersion in their collapse energies while the skyrmion stability is strongly correlated with its size with a power-law dependence. Topological solitons above the critical point in yellow show large collapse energy barriers and their energy at a given size is tunable. When the magnetic field increases, these solutions show the same power-law collapse energy versus radius dependence as skyrmions. Topological solitons and skyrmions of 20 nm and above present collapse barriers larger than $21 k_B T_{293K}$, which corresponds to lifetimes longer than 1s (Arrhenius-Néel law with a try rate $1/\tau_0 = 10^9$ Hz) and their stability increases with size and can further be increased by parameter engineering (see also [46]).

## 7  Conclusion

We have developed an analytical topological soliton model containing expressions of the long range demagnetising and exchange curvature energies, two key ingredients to stabilize bubbles and skyrmions in ferromagnetic thin films. This allowed us to study systematically topological soliton solutions over a wide range of parameters and explore quantitatively the possible transitions between small and large topological solitons. The observed skyrmion-bubble transition present similarities with the liquid-gas transition, in particular a critical point is present above which the transformation between both spin textures becomes continuous. While distinct characteristics of skyrmions and bubbles remain, their common nature as topological solitons is emphasised. Above the critical $(D_{cs}, H_{cs})$ point, the topological soliton can not be strictly named a skyrmion or a bubble, as it possesses some characteristics of both spin textures. This hybrid between a bubble and a skyrmion may be referred to as a supercritical skyrmion.

## Acknowledgements

A.B-M thanks Alex Kruchkov for discussions at the beginnig of the project, Boris Ivanov for valuable explanations on 2D topological solitons, Nikolai Kieselev for detailed comments on the manuscript. Special thanks to André Thiaville, Stanislas Rohart, Olivier Fruchart for discussion on the demagnetising field. Thanks to Albert Fert, Vincent Cros and Henrik Rønnow for their support.

**Author contributions**   A.B-M conceived the original idea. A.B-M, L.C., A.W. and L.R. worked on the analytical model. A.B-M, L.R., N.R. and M.D. carried out numerical calculations. A.B-M wrote the paper. A.B-M, L.C., A.W. and L.R. participated to the corrections of the paper.

**Funding information**   This work was supported by the Centre National de la Recherche Scientifique (CNRS), the French National Research Agency (ANR) under the project ELEC-SPIN ANR-16-CE24-0018 and the Laboratoire d'Excellence (LabEx) Laboratoire d'Alliances Nanoscience-Energies du futur (LANEF).

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
