# Peer review of "The skyrmion-bubble transition in a ferromagnetic thin film"

_SciPost Physics, doi:SciPost Phys. 4, 027 (2018)_

## Round 2 · Referee Report · Anonymous (Referee 1) · 2018-1-8

Strengths

  1. If the weak points could be addressed, the paper would be useful to the experimental community in their search for skyrmions in DM materials, and in guiding the fabrication of new such materials.

  2. The presentation is clear and detailed.

Weaknesses

  1. Expression (9) is usually a good approximation for the energy, but it is not easy to know when this could break down. Despite the detailed explanations in the paper on the limitations of the approximation, there may be further unexpected cases that this breaks down.

  2. Α weak point of the paper is that the results are not compared against simulations. It would be desirable that the subtle regions of the phase diagram (the region of skyrmion and bubble having similar radii or merging, and the regions of skyrmion bursting, or bubble collapsing) be confirmed by simulations. For example, it appears, in the phase diagram in Fig. 3f, that the bubble exist almost down to external field H=0. As far as I know, existing numerical simulations do not confirm this. Another example is zone 0 in Fig. 3b. It should be easy to check numerically whether this approximate result carries over to the full model.

  3. Apart of all the above comments, the relation of the present paper to previous works should be clarified. (Indeed, a very good presentation of the literature is provided). Specifically, Ref. [46] claims to provide a "Full phase diagram of isolated skyrmions". Apart from differences in the presentation of the phase diagram, I cannot see what is the additional information provided in the present paper compared to the results presented in Ref. [46]. I would ask the authors to clarify this.

Report

Magnetic skyrmions in Dzyaloshinskii-Moriya (DM) materials is a very active field of research in the last years and the perspectives for applications is supporting research activity. Despite the quantity of related research papers the understanding of a skyrmion configuration is not yet complete. Specifically, the community is not yet confident about the relation of a skyrmion (in DM materials) to a magnetic bubble (in perpendicular anisotropy materials, known since the 1960s).
The present paper is a contribution towards a quantitative understanding of the relation between skyrmions and bubbles and it provides a phase diagram for the two states.

The theoretical model for the energy of an axially symmetric skyrmion in a DM material is given in Eq. (9). The terms come from obvious approximations of the (a) exchange and anisotropy energy for a domain wall, (b) DM interaction, and (c) Zeeman energy, while the magnetostatic energy is given by an approximation that appears in Ref. [47].
The rest of the paper is a study of this expression as a function of the skyrmion radius, while the DM parameter D and the external field H are parameters that may be varied.
Based on previous work on skyrmions and on work on bubbles, it can easily be anticipated that this function may have two minima.
It is interesting to follow these minima. The paper explains the cases that these are well separated and the cases that they merge to a single minimum. In the later case a skyrmion and a bubble is one and the same thing.

Requested changes

  1. Eq. (5) is claimed to give a good fit "without adding any fitting parameters". Actually all numbers appearing in Eq. (5) are fitting parameters, that have already been given appropriate values.

  2. Eq. (8) is explained to give "demagnetising energy difference between a magnetic cylinder with uniform magnetisation pointing in one direction and the uniform ferromagnetic state". This is not a precise expression. What is probably meant by "cylinder" is a uniformly magnetized film with a cylindrical domain with inverted magnetization?

  3. In the beginning of Sec. 3 it is stated that "changes in the fixed parameters do not modify qualitatively the results presented here but rather shift the main features to different D values." This result would probably follow trivially if one would work in a nondimensional form of the equation.

  4. All the information in Figs. 1a,b,d,e seems to be contained in Fig. 1f. What is there in a,b,d,e that is not contained in Fig. 1f? All the information in Figs. 1c seems to be contained in Fig. 1g. If this is correct then Fig. 3 need to contain only entries f and g.

  5. In the figure captions (e.g., Fig. 3 and 4) it is, in same cases, not clear which entry the description is referred to. It would be clearer if each entry is explained separately.

  6. In many places the expression "spin rotation energy" is used. This expression is not a clear one.

  7. In Sec. 4.1.1 (and in various other places) there is the expression "a blue dot in Fig. 3b above which...". Referring to a "region" above a "point" is not a clear specification.

  8. In Sec. 4.1.2 we read "Zone 1 starts along the blue line". It is not clear what is meant.

  9. In Sec 2.4.4 and 4.3.2 it is claimed that errors are negligible because some energy term is small compared to other terms. But, comparing energies is not safe. What matters is rather compare variations of energies.

  • validity: high
  • significance: good
  • originality: ok
  • clarity: high
  • formatting: perfect
  • grammar: excellent

Author:  Anne Bernand-Mantel  on 2018-04-09  [id 238]

(in reply to Report 1 on 2018-01-08)

We thanks the referee for his valuable comments and his interest for our work. Here is our response to the weaknesses and request changes as well as a list of changes. The references mentioned with a number in brackets correspond to the version of the paper read by the referee (v2). A revised version has been submitted to arxiv (v3).

Referee : 1.”Expression (9) is usually a good approximation for the energy, but it is not easy to know when this could break down. Despite the detailed explanations in the paper on the limitations of the approximation, there may be further unexpected cases that this breaks down.”
Authors: As mentioned by the referee we have discussed in details the limitations of the model. First, it is restricted to films thinner than the exchange length and to DMI values sufficiently large to ensure a Néel nature of the DW. In addition, the approximations used will have an impact on the estimation of the skyrmion and bubble equilibrium radius mainly due to the approximations on the long range demagnetizing energy and on the Zeeman energy. This error, as described in the text, is small in the r<$\Delta$ region, as both of these energies and energies variations are small, compared to the other energies and energies variations, in this range. The error is also negligible in the r>>$\Delta$ range as the demagnetizing energy and the Zeeman energy are well-approximated in this range. In the intermediate radius range the error on the topological soliton radius may be significant in particular if $\Delta$ is large (weak anisotropy) and if we are close to the critical (Dcs,Hcs) point where the radius presents a strong susceptibility to energy variations. Despite this potential errors in the topological soliton radius estimation, the general features of the skyrmion-bubble diagram described un the paper will not be modified, as confirmed by our complementary micromagnetic simulations that we have carried out close to the critical (Dcs,Hcs) point (see description in pargraph 2. below).

Referee : 2. “Α weak point of the paper is that the results are not compared against simulations. It would be desirable that the subtle regions of the phase diagram (the region of skyrmion and bubble having similar radii or merging, and the regions of skyrmion bursting, or bubble collapsing) be confirmed by simulations.”
Authors: We have carried out complementary numerical simulations, which are presented in the new section 4.4. The micromagnetic phase diagram close to (Dcs,Hcs) showed in Figure 4a can be compared to the phase diagram obtained with the analytical model in Figure 4b. The results are very similar to what we predict with our analytical model using the same parameters, but we observe a shift towards 8% larger D values as expected and discussed in section 4.3.1. This confirms that the approximations in our model leads to an overestimation of Dcs, however all the predicted behaviors are qualitatively the same. We would like to point out that the diagram in Figure 4a realized using Mumax3 took days to be calculated while the diagram in Figure 4b is calculated in milliseconds. In conclusion, we think that micromagnetic and analytical models are complementary approaches.
Referee: “For example, it appears, in the phase diagram in Fig. 3f, that the bubble exist almost down to external field H=0. As far as I know, existing numerical simulations do not confirm this."
Authors: While our topological soliton model shows that there is a solution almost down to H=0, we introduce in section 1.1 a limit in our model visible as a dashed line in figures 3 a,b,d and f. As described in section 4.1.1 this line correspond to the limit where the observation of an isolated topological soliton is very unlikely as the uniform ferromagnetic state is not the ground state anymore and stripe phase will appear (see also Kiselev. ref. 35). This may explains why bubbles at low magnetic field are not observed in the numerical simulations mentioned by the referee.
Referee: "Another example is zone 0 in Fig. 3b. It should be easy to check numerically whether this approximate result carries over to the full model. “
Authors: The absence of skyrmion for too low values of D has been observed in previous numerical simulations (Sampaio Nat. Nanotech. 8, 839 (2013), Rohart PRB 88, 184422 (2013),…)

Referee: 3. “Apart of all the above comments, the relation of the present paper to previous works should be clarified. (Indeed, a very good presentation of the literature is provided). Specifically, Ref. [46] claims to provide a "Full phase diagram of isolated skyrmions". Apart from differences in the presentation of the phase diagram, I cannot see what is the additional information provided in the present paper compared to the results presented in Ref. [46]. I would ask the authors to clarify this. “
Authors: We agree that our work is closely related to Ref.[46] from Büttner et. al. (Scientific Reports 8, 4464 (2018)), this is why we mention their work several times in the paper. Both of our model and the model from Büttner et. al. are analytical models which aims at describing all energy terms for the stabilization of skyrmions and bubbles. However, there are strong difference between both models. The analytical model from Büttner et. al., which is avaliable in their supplementary materials, involves the use of a large number of fitting constants and complicated fitting functions which present artificial non monotonous variations as attested by the fluctuations of the errors presented in figures S1-S4 in the supplementary. Considering this, it is difficult to say if the multiple extrema observed by Bütter et. al. are not artifacts of the model. Another major difference between our model and the model from Büttner et al. is that their model present divergence or strong errors which makes it unusable in the r/$\Delta$ < 0.88 range while our model does not present any divergence and can be used down to r/$\Delta \rightarrow$ 0. For example, in the model from Büttner et. al. the exchange energy described in supplementary S2.2 possess a 1/$\rho$ term which diverges for $\rho \rightarrow 0$ and the anisotropy energy becomes negative for $\rho \rightarrow 0$. Despite these issues, some main features predicted by our model are also present in the phase diagram from Büttner et. al. (Figure 2.c): above a D value close to 2 mJ/m$^2$ continuous transitions between bubbles and skyrmions are observed and below this value, discontinuous transition and a region of bistability is present. These features are very similar to what we report.

Requested changes
Referee: 1. “Eq. (5) is claimed to give a good fit "without adding any fitting parameters". Actually all numbers appearing in Eq. (5) are fitting parameters, that have already been given appropriate values.”
Authors: We agree with the referee and we have suppressed this sentence. (see list of change 1)
Referee: 2. “Eq. (8) is explained to give "demagnetising energy difference between a magnetic cylinder with uniform magnetisation pointing in one direction and the uniform ferromagnetic state". This is not a precise expression. What is probably meant by "cylinder" is a uniformly magnetized film with a cylindrical domain with inverted magnetization?”
Authors: We thanks the referee for this remark we have corrected the sentence. (see list of change 2)
Referee: 3. “In the beginning of Sec. 3 it is stated that "changes in the fixed parameters do not modify qualitatively the results presented here but rather shift the main features to different D values." This result would probably follow trivially if one would work in a nondimensional form of the equation.”
Authors: We agree with the referee, we wanted to work with dimensional parameters for two reasons : 1) to make our results directly readable for experimentalist who are familiar with the parameters we use which are typical of thin films with perpendicular anisotropy. 2) as our programs are available to the reader it is easy to run the simulations with different parameters.
Referee: 4. “All the information in Figs. 1a,b,d,e seems to be contained in Fig. 1f. What is there in a,b,d,e that is not contained in Fig. 1f?
All the information in Figs. 1c seems to be contained in Fig. 1g.
If this is correct then Fig. 3 need to contain only entries f and g. »
Authors: We are not sure about which figures the referee refers to as there is no Figure 1f nor 1g. We agree that some informations are represented in two different ways in some figures for example in Fig. 1d and e. The aim of Fig. 1d is to emphasis the deviation of $\sigma_s$ compared to $\sigma_w$ due to the bending of the domain wall in the (x,y) plane.
Referee: 5. “In the figure captions (e.g., Fig. 3 and 4) it is, in same cases, not clear which entry the description is referred to. It would be clearer if each entry is explained separately.”
Authors: We have improved the figure caption (see list of change 3)
Referee: 6. “In many places the expression "spin rotation energy" is used. This expression is not a clear one.”
We agree this point was unclear. We clarified this. (see list of change 4-7)
Referee: 7. “In Sec. 4.1.1 (and in various other places) there is the expression "a blue dot in Fig. 3b above which...". Referring to a "region" above a "point" is not a clear specification.”
Authors: We clarified this. (see list of change 8)
Referee: 8. “In Sec. 4.1.2 we read "Zone 1 starts along the blue line". It is not clear what is meant.”
Authors: We clarified this. (see list of change 9)
Referee: 9.” In Sec 2.4.4 and 4.3.2 it is claimed that errors are negligible because some energy term is small compared to other terms. But, comparing energies is not safe. What matters is rather compare variations of energies.”
Authors: We agree, we have corrected this. (see list of change 10)

Author:  Anne Bernand-Mantel  on 2018-04-20  [id 246]

(in reply to Anne Bernand-Mantel on 2018-04-09 [id 238])

This is a reply to the anonymous comment on 2018-04-13

The volume stray field is negligible in the ultrathin limit we consider. Concerning surface stray fields they are not included only in the limit r≫Δ. We use a single model to calculate the energy at all scales, this model contains a local stray field energy term (see section 2.2) which is the dominating stray field energy for low radius skyrmions and a long range stray field energy term (see section 2.4.4) which is negligible compared to the total energy and energy variations for low radius skyrmions but becomes important in the r≫Δ regime. Despite the simplicity of this model it allows us to capture the phenomenon of the skyrmion-bubble transition: the bistability between skyrmions and bubbles solutions and the existence of continuous and discontinuous transitions between these two objects, as also predicted by Büttner et. al. Sci. Rep 2018. This is also confirmed by micromagnetic calulations, as described in section 4.4).

Anonymous say that the key difference between the two theories is that Büttner's model can predict skyrmion energies in multilayers up to 100 nm. However for thicknesses between 10 and 100 nm it is unlikely that the magnetization does not vary in the z-direction, as assumed by their theory, and the reorientation of DW close to the surface will appear to lower the demagnetizing energy even in presence of significant DMI. This has been demonstated experimentally and theoretically in the work of Legrand et. al. (arXiv:1712.05978v2) which consider multilayers very close to the system described by Bütter et. al..

Anonymous on 2018-04-13  [id 241]

(in reply to Anne Bernand-Mantel on 2018-04-09 [id 238])
Category:
correction

This comment is a response to the following statement:

"We agree that our work is closely related to Ref.[46] from Büttner et. al. (Scientific Reports 8, 4464 (2018)), this is why we mention their work several times in the paper. Both of our model and the model from Büttner et. al. are analytical models which aims at describing all energy terms for the stabilization of skyrmions and bubbles. However, there are strong difference between both models. The analytical model from Büttner et. al., which is avaliable in their supplementary materials, involves the use of a large number of fitting constants and complicated fitting functions which present artificial non monotonous variations as attested by the fluctuations of the errors presented in figures S1-S4 in the supplementary. Considering this, it is difficult to say if the multiple extrema observed by Bütter et. al. are not artifacts of the model. Another major difference between our model and the model from Büttner et al. is that their model present divergence or strong errors which makes it unusable in the r/Δ < 0.88 range while our model does not present any divergence and can be used down to r/Δ→0. For example, in the model from Büttner et. al. the exchange energy described in supplementary S2.2 possess a 1/ρ term which diverges for ρ→0 and the anisotropy energy becomes negative for ρ→0. Despite these issues, some main features predicted by our model are also present in the phase diagram from Büttner et. al. (Figure 2.c): above a D value close to 2 mJ/m$^2$ continuous transitions between bubbles and skyrmions are observed and below this value, discontinuous transition and a region of bistability is present. These features are very similar to what we report."

There are several mischaracterizations in these sentences. First, the energy functions presented by Bernand-Mantel et al. are indeed simpler than what is derived in Ref. [46]. However, this comes at a high price: volume stray fields are not included at all in the presented model and surface stray fields are only included in the limit $r\gg\Delta$. Therefore, the transition between bubble-like ($r\gg\Delta$) and point-like ($r\approx \Delta$) skyrmions is not captured well by the presented model. The only reason why there is reasonable agreement between the model and simulations in this manuscript is that the film thickness is chosen to be t=0.5 nm, i.e., small enough to make non-local stray fields very weak. In the experimentally much more relevant case of multilayers with several nanometers of thickness (up to 100 nm), the approximations in the presented model give significant errors. This is the key difference between the two theories.

The second misunderstanding is the treatment of the small $r$ limit. The theory in Ref. [46] can describe arbitrarily small skyrmions. That paper just uses a different definition of $\Delta$ than Bernand-Mantel et al., which leads to the misstatements in the comments by Bernand-Mantel et al. In Ref. [46], the domain wall width $\Delta$ is a dynamic parameter that is linked to the inverse of the slope of $m_z$ near $m_z=0$, and as such depends on r (which is the zero-crossing of $m_z$). For a bubble-like skyrmion, these parameters are independent and $\Delta$ is indeed equivalent to the straight wall width (which, by the way, also deviates from $\sqrt{A/K_\text{eff}}$ for intermediately thick films, see Ref. [39]). Hence, the condition $r>0.88 \Delta$ simply reflects that for point-like skyrmions, the radius $r$ (zero-crossing) is proportional to the inverse slope $\Delta$, which is a strict mathematical requirement for any $m_z$ profile that complies with $m_z(\rho=0)=1$.

The suggestion that the exchange energy term diverges and the anisotropy energy term can become negative in the expressions presented in Ref. [46] is based on the misunderstanding outlined above. The well-defined and accurate behavior of the exchange and anisotropy expressions in that paper is clear from Figs. S1a and S2a in the Supplemental Information for Ref. [46].

Finally, the model in Ref. [46] does predict some tiny energy oscillations for some choices of parameters that look like unphysical local extrema. However, the depth of these extrema is is much smaller than kT. That is, stable minima are always qualitatively and quantitatively accurate.

---

## Round 3 · Referee Report · Anonymous (Referee 2) · 2018-4-13

Strengths

1. The paper would be useful to the experimental community in their search for skyrmions in DM materials, and in guiding the fabrication of new such materials.

2. The presentation is clear and detailed.

3. The analytical model studied in this paper while being particularly simple, it does capture the features of a complicated phenomenon.

Report

The analytical model studied in this paper, while being particularly simple, it does capture the features of a complicated phenomenon. That is probably the main strength of the paper.

The resubmitted version of the paper removes the weak points. I list below some minor comments on the changes submitted with the new version.

Weakness point 2:
It is an improvement of the paper that the authors have added results of numerical simulations and discussed the comparison with their analytical model.
This has verified, for example, that the details about the point Dcs, Hcs are given correctly qualitatively and to a fine approximation quantitatively.
I would like to make clear here that I actually considered there was no doubt that the overall picture from the analytical model is correct, thus I did not think that a full set of simulations would be necessary.
In any case, the authors have clarified the details of the phase diagram.

Weakness point 3:
The explanations of the authors' reply regarding the relation of their work to previous works is sufficient. I would consider it though better if the authors had included some of these explanations in the paper.
In any case, I understand that the Reports and the Replies are accessible, just like the paper itself, so this information will be available.

Requested change 4:
This was referring to Fig. 3 (not to Fig. 1 as I erroneously wrote).
I still think some of the entries are redundant. However, it is up to the authors to decide whether these rather help make their results clearer.

---

## Round 3 · Author Response

Dear Editor,
To answer to the referee comments, we have modified the manuscript as described in the list of changes. In addition to minor corrections, we have carried out complementary micromagnetic simulations wich has been included in the manuscript in section 4.4.
Anne Bernand-Mantel

---

## Round 3 · List of Changes

list of changes:
1. (requested change 1) We have replaced the sentence “This expression reproduced the exchange energy obtained in Section 2.3 with a 3% maximum error without adding any fitting parameters.” By “This expression reproduce the exchange energy obtained in Section 2.3 with a 3% maximum error in the full radius range.”
2. (requested change 2) We have replaced the sentence “ We use an approximate expression for the Zeeman energy of the topological soliton which represents the Zeeman energy difference between a film with a uniform +Ms containing a magnetic cylinder of radius rand opposite uniform -Ms magnetisation, and the Zeeman energy of the uniform +Ms state.
3. (requested change 5) The beginning of the caption of figure 3 has been replaced.
4. (requested change 6) We have replaced the expression “spin rotation energy” by “topological soliton wall energy” which is defined in section 2.3 as “the topological soliton wall energy density is defined as the sum of the exchange, anisotropy and DMI energy densities”
5. (requested change 6) In section 2.4.2 we have suppressed “and is decreasing the energy cost of the spin rotation”
6. (requested change 6) Description of the total energy in section 2.4.5.
7. (requested change 6) “spin rotation energy” has been replaced by “topological soliton wall energy” in several places.
8. (requested change 7) We have replaced “above which” by “at which”. We have suppressed “is starting at the critical point (Dcs,Hcs) and”.
9. (requested change 8) We have replaced “Zone 1 starts along the blue line” by “Zone 1 starts for D values larger than D_{\mathrm{min.}}=2\sqrt{A_{\mathrm{ex.}}K_{\mathrm{eff.}}}/”
10. (requested change 9) We have suppressed the sentences in Sec 2.4.4: “However, is limited as in this radius range Elong rangedem. is at least one order of magnitude smaller in amplitude than Eexch., EDMI and Eanis. (see Fig.???e) as also shown by B\"uttner et al. \cite{Buttner2017}. “ and modified the sentence in Section 4.3.2 “the Zeeman energy and its variations are always one to several orders of magnitude smaller than the total energy value and total energy variations in the r<Δ range.”
11. We have added the section 4.4 with a new figure where the micromagnetic simulations are presented

---

## Editorial Decision

published